

# Brief Communication: Meteorological and climatological conditions associated with the 9 January 2018 post-fire debris flows in Montecito and Carpinteria California, USA

**Nina S. Oakley[1,2], Forest Cannon[2], Robert Munroe[3], Jeremy T. Lancaster[4], David Gomberg[3], F. Martin Ralph[2]**

[1]Western Regional Climate Center, Desert Research Institute, 2215 Raggio Parkway, Reno, Nevada, USA, 89512

[2]Center for Western Weather and Water Extremes, Scripps Institution of Oceanography, 9500 Gilman Dr., La Jolla, CA, USA, 92093

[3]National Weather Service, Oxnard/Los Angeles, 520 N. Elevar St., Oxnard, CA, USA, 93030

[4]California Geological Survey, 801 K St., Sacramento, CA, USA, 95814

*Correspondence to*: Nina S. Oakley (nina.oakley@dri.edu)

**Abstract.** The Thomas Fire burned 114,078 hectares in Santa Barbara and Ventura Counties, southern California, during
December 2017-January 2018. On 9 January 2018, high intensity rainfall occurred over the Thomas Fire burn area in the mountains above the communities of Montecito and Carpinteria, initiating multiple devastating debris flows. The highest rainfall intensities occurred with the passage of a narrow rainband along a north-to-south oriented cold front. Orographic enhancement associated with moist southerly flow immediately ahead of the cold front also played a role. We provide an explanation of the meteorological characteristics of the event and place it in historic context.

**1 Introduction**

The Thomas Fire was ignited on 4 December 2017 and burned 114,078 hectares in Santa Barbara and Ventura Counties in southern California before it was 100% contained on 12 January 2018. It became the largest wildfire in California's modern history. Soil burn severity was predominately moderate with small areas mapped as high in the northern and western portions of the burn area (CAL FIRE 2018). In combination with the steep terrain and underlying geology, the
United States Geological Survey (USGS) rated watersheds north of the Santa Barbara coastal plain and Ojai as having high debris flow hazard based on a design rainstorm of 15-minute rainfall intensity of 24 mm h$^{-1}$ (USGS 2018a; Fig. S1).

On 9 January 2018 between 11:30-12:00 UTC (3:30-4:00 LST), high intensity rainfall occurred over the westernmost portion of the Thomas Fire burn area, exceeding the USGS 15-minute design storm by more than threefold at some locations. Large magnitude debris flow surges were triggered in multiple watersheds, overwhelming debris basins and
issuing onto urbanized alluvial fans including the communities of Montecito and Carpinteria (Fig. 1). The debris flows were



devastating, resulting in 23 deaths, 246 structures destroyed and 167 damaged (County of Santa Barbara 2018). Preliminary loss estimates for residential and commercial property alone have exceeded $421 million USD (California Dept. of Insurance 2018).

5 Over the past three decades, more than a dozen notable post-fire debris flow (hereafter "PFDF") events have been observed across the Transverse Ranges of southern California (Oakley et al. 2017), where steep terrain, highly erodible soils, and frequent wildfires create favorable conditions for PFDFs (Wells 1987). In the Montecito area specifically, damaging PFDFs last occurred following both the Coyote Fire of 1964 and Romero Fire of 1971 (U.S. Army Corps of Engineers 1974).

10 This manuscript describes the meteorological origins of the high intensity precipitation leading to the 9 January 2018 debris flow and places the event in a climatological context. This information is intended to appeal to a broad community of researchers and stakeholders to support investigations on this and other PFDFs in a range of fields such as geomorphology, social science, atmospheric modeling, and pollutant transport, and to increase situational awareness for future events.

## 2 Meteorological analyses

### 15   2.1 Synoptic conditions

The large-scale meteorological conditions that led to the development of intense precipitation and the debris flow event (12 UTC 9 January) featured an upper level (500 hPa) closed low-pressure system centered slightly offshore of Point Conception at approximately 34° N, 122° W (Fig. S2). The attendant 1000 hPa surface low-pressure center was situated slightly further north and east, at approximately 36° N, 121° W (Fig. 2a), and had continued to deepen preceding the PFDF 20 event. Integrated vapor transport (IVT) exceeding 400 kg m$^{-1}$ s$^{-1}$ was observed on the eastern side of the large-scale circulation, over the Southern California Bight (Fig. 2, S4). This moisture plume resulted from the re-organization of remnant moisture from an atmospheric river (a subtropical moisture plume; AMS 2018) that had propagated across the region the previous day and was present over central Baja California, Mexico, at the time of the event (Fig. 2a). Integrated water vapor (IWV) values in the flow impinging on the Santa Ynez Mountains at the time of the PFDF event were 25 approximately 30 mm (Fig. S5). The IWV and IVT values associated with this event are consistent with a weak atmospheric river.

At the time of the event, the downstream side of a cyclonically-curved upper level (250 hPa) jet was located over southern California with a 50 m s$^{-1}$ jet streak exit region situated over the Santa Barbara area (Fig. 2a; S3). Model soundings indicate cold air advection between approximately 450-600 hPa, increasing the lapse rate in this layer and creating a region 30 of potential instability (Fig. S6). This was collocated with a region of inferred absolute vorticity advection by the geostrophic wind at 500 hPa (Fig. S2). These conditions were also associated with a well-defined cold front parallel to and impinging





upon the moist low-level jet ahead of the front (Fig. 2). This scenario positioned the Thomas Fire burn area in a favorable region for large-scale ascent and destabilization of the atmosphere (Markowski and Richardson, 2010).

**2.2 Mesoscale conditions**

At approximately 9 UTC 9 January, the north-to-south oriented cold front was located just offshore of Point Conception and was propagating eastward across the Southern California Bight (Fig. 2b). Radar and satellite imagery reveal a narrow band of intense rainfall and vigorous convection parallel to and in the vicinity of the cold front (Fig. 3, S7), a feature known as a Narrow Cold Frontal Rainband (NCFR; Markowski and Richardson 2010). These features tend to form when there is strong convergent flow along the surface front and divergent flow aloft. This facilitates the release of potential instability through forced ascent and the generation of intense convective precipitation bands (Hobbs and Persson, 1982; Markowski and Richardson 2010).

Ahead of the cold front, 10-20 ms$^{-1}$ southeasterly winds were present below 1 km, as demonstrated in measurements from the 449 MHZ radar wind profiler situated at Santa Barbara Airport (SBA; Fig. S8). The cold front passed over Santa Barbara and Montecito between 11:00 and 12:00 UTC. The passage of the cold front can be observed as a shift from strong southeasterly winds to weak south to southwesterly winds below 2 km (Fig. S8). The convergence of this southeasterly flow ahead of the front and westerly flow behind the front helped support the development of the NCFR. As the cold front made landfall and encountered the complex coastal terrain, the NCFR became segmented and dissipated in some areas. One segment became well organized north of Santa Rosa and Santa Cruz Islands and intensified as it moved across the Santa Barbara Channel towards Montecito (Fig. 3b).

Strong, low-level south-to-southeasterly winds peaking near 1 km were observed immediately ahead of the cold front, a feature known as a low-level jet (LLJ; Neiman et al. 2004; Fig. 2b, S8). The presence of the LLJ orthogonal to the terrain combined with available moisture (Fig. 2b, S8) creates a situation favorable for orographic precipitation enhancement (Lin et al. 2001). However, due to the forcings described and presence of high radar reflectivity values over the ocean before the NCFR impacted the terrain (Fig. 3b). it appears that the NCFR was the dominant feature producing short-duration high intensity rainfall in this case, with orographic forcing likely acting as a complementary, but secondary, factor.

Radar and surface-based precipitation observations reveal that the segment of the NCFR impacting Montecito and Carpinteria began to dissipate near the Santa Barbara-Ventura County line just after 12 UTC (4:00 AM PST; Fig. 1, S9). The subsequent weakening of rainfall intensity likely spared other portions of the burn area from additional catastrophic debris flows.

**3 Historical and climatological context**

**3.1 Historical context of precipitation event**



The Santa Barbara County Public Works Department (SBCPWD) maintains a network of precipitation gauges used for flood hazard and water resource management, with records dating back to the 1960s. All precipitation data discussed herein have been archived and quality controlled by SBCPWD and can be accessed at: www.countyofsb.org/pwd/hydrology.sbc.  Average return intervals (ARI) described in this section are sourced from NOAA

5    Atlas 14 (hdsc.nws.noaa.gov/hdsc/pfds/; Bonnin et al. 2006) for the coordinates of each station. Tables S1-S3 provide further information on relevant observations in and around the Thomas Fire burn area.

The short duration intense precipitation (Fig. 1, 3c) observed during the 9 January 2018 debris flow event was exceptional and in some cases broke individual station records, but was not unprecedented for Santa Barbara County. At the 5-minute duration, a maximum of 15.24 mm was recorded at Jameson Dam, a 25-year ARI event (25-1000 year at 90%

confidence). This exceeded the previous record 5-minute observation of 13.46 mm set in Water Year (WY; September-August) 1998. Records considered at Jameson Dam began in 1965. At the Carpinteria Fire Station (FS), the maximum 5-minute precipitation observation was 9.14 mm, a 10-year ARI event (10-25 year at 90% confidence). This observation tied the existing record set in WY1969. Records considered for Carpinteria FS began in 1964. In Montecito, 13.72 mm was observed in 5 minutes, setting a record for this station, though the station record is very short, beginning in 2009. This

registers as a 200-year ARI event (100-1000 year at 90% confidence).

At the 15-minute duration, Jameson Dam set a record of 25.15 mm, exceeding the previous record of 13.46 mm set in WY1998. The Carpinteria FS gauge also set a record at this duration, observing 21.84 mm and defeating the previous record of 16.76 mm recorded in WY1975. Additionally, a 15-minute record of 18.54 mm was set at the Montecito station. At Jameson, this is a 25-year ARI event (10-500 year 90% confidence) and for Carpinteria FS and Montecito, this is a 50-year

ARI event (25-1000 year at 90% confidence).

The records set at individual stations at the 5- and 15-minute durations were well shy of the extremes observed in Santa Barbara County. At the 5-minute duration, the County record is 18.29 mm at the UCSB station set in WY1998. The County record at the 15-minute duration is 35.31 mm set at the San Marcos Pass station in WY2015.

At the 1-hour and longer durations, precipitation intensities were generally less than the 10-year return interval.

Storm total precipitation over a 24-hour period was roughly 50-75 mm at low elevations and 100-125 mm at higher elevations (Fig. 3d). These 24-hour precipitation totals were mostly less than the one-year return interval.

**3.2 Context of intense rainband**

No known documentation exists on the abundance of NCFRs or similar frontal convection in southern California,

though several resources acknowledge their occurrence and impacts. We hypothesize that these features occur multiple times in a given year and are not uncommon in landfalling atmospheric rivers with strong cold fronts impacting the region. NCFRs such as the one observed during the 9 January 2018 event (Fig. 3a, b) have been previously associated with post-fire debris flows in the Transverse Ranges of southern California.



On 12 December 2014, an NCFR produced intense rainfall over the Springs Fire burn area in Camarillo, CA, initiating a debris flow that destroyed several homes (Fig. S10; Sukup et al. 2016; Oakley et al. 2017). More recently, on 20 January 2017, a narrow band of high intensity rainfall occurring along a cold front produced a debris flow on the Sherpa Fire burn area in western Santa Barbara County (Fig. S11). Five cabins and over 20 vehicles were damaged in El Capitan

Canyon, and nearly two-dozen people had to be rescued (Lin et al. 2017). Neiman et al. (2004) used observations from an intensive field campaign in 1998 to detail the synoptic and mesoscale forcing associated with a cold front that also generated high-intensity precipitation in this region of Southern California. Observations of convective precipitation bands in the area and their precipitation impacts date back to the early 1960s (Elliot and Hovind 1964). However, NCFRs are not the only mechanism for producing high intensity precipitation and post-fire debris flows in Southern California. Thunderstorms and

orographically-forced precipitation have historically resulted in post-fire debris flows as well (Oakley et al. 2017).

Intense precipitation associated with NCFRs is not unique to southern California and commonly occur in other parts of the world where there is also complex terrain. These features have been observed to impact Chile (Viale et al. 2013) in South America, and Western Europe (Roux et al. 1993; Gatzen 2011). These areas may experience severe wildfires, and NCFRs may serve as PFDF triggers in these regions as well.

### 4 Conclusion

The Transverse Ranges of southern California are prone to post-fire debris flows. Following a wildfire of moderate to high burn severity on steeply sloping terrain, short-duration, high intensity precipitation over these areas may trigger a debris flow, as occurred on the morning of 9 January 2018.

The debris flows were triggered by a band of intense precipitation along a cold front, known as a narrow cold frontal rainband (NCFR; Fig. 2b and 3a,b) that impacted the westernmost portion of the Thomas Fire burn area. Such rainbands develop due to vertical circulations along the front that facilitate low-level convergence and lifting, which can force convection and intense rainfall. This mesoscale process may also benefit from destabilization at large-scales through the inferred synoptic forcing for ascent.

A weak atmospheric river was present at the time of the event, facilitating moisture availability. Conditions along and ahead of the cold front were favorable for orographic enhancement in the Santa Ynez Mountains above Montecito and Carpinteria. Observations suggest that the NCFR, and to a lesser extent, orographic forcing produced high intensity rainfall in this event. This demonstrates the value of examining both mesoscale and synoptic conditions when forecasting short-duration, high intensity precipitation associated with mid-latitude cyclones.

Precipitation in this event was extreme at the 5- and 15-minute durations; two locations recorded >13 mm in 5 minutes. Records were set at a few stations with 50+ years of observations. However, the intensities observed are not unprecedented for Santa Barbara County. Storm total precipitation was unremarkable for the area, with 24-hour totals at the <1-year return interval, demonstrating that significant storm total precipitation is not required to produce rainfall capable of initiating a post-fire debris flow.




This analysis supports improved situational awareness and understanding of rainfall events producing post-fire debris flows for the natural hazards community in California and other mid-latitude regions of the world that experience wildfires in complex terrain. It also serves as a meteorological description to inform research on a variety of topics related to the 9 January 2018 debris flows. Future work will examine the role of terrain in modifying the NCFR and precipitation processes described herein, the predictability of forced convection and extreme precipitation rates, and their relation to post-fire debris flows in other regions of the world.

**Data Availability**

All data can be accessed at the web addresses provided within the text or references.

**Acknowledgements**

This material is based upon work supported by the U.S. Geological Survey under Cooperative Agreement No. G16AC00266 for the Southwest Climate Adaptation Science Center.

**Competing Interests**

The authors declare they have no conflict of interest.

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



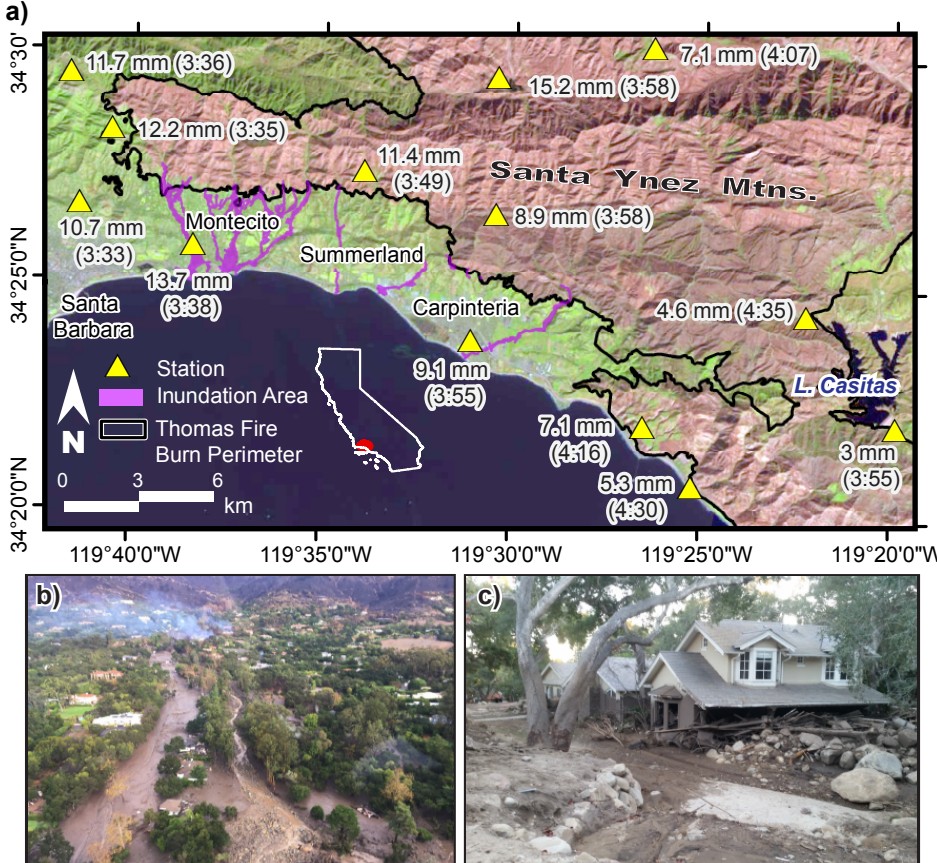

**Figure 1: a) Burned areas within the perimeter of the Thomas Fire are depicted in red and unburned areas in green, as derived from the Landsat 8 thermal infrared sensor and surface reflectance imagery (USGS 2018b). Inundated areas, as mapped by the California Geological Survey, are displayed in purple. Station-based observations of greatest event 5-minute precipitation and the**
5  **start time of the interval (LST) are labeled. b) Aerial photo of San Ysidro Creek in Montecito following the debris flow; areas that were once roads and homes appear as rivers of mud and debris. Photo: Ventura County Air Unit. c) A home destroyed in the debris flow. Photo: Brian Swanson, CGS.**



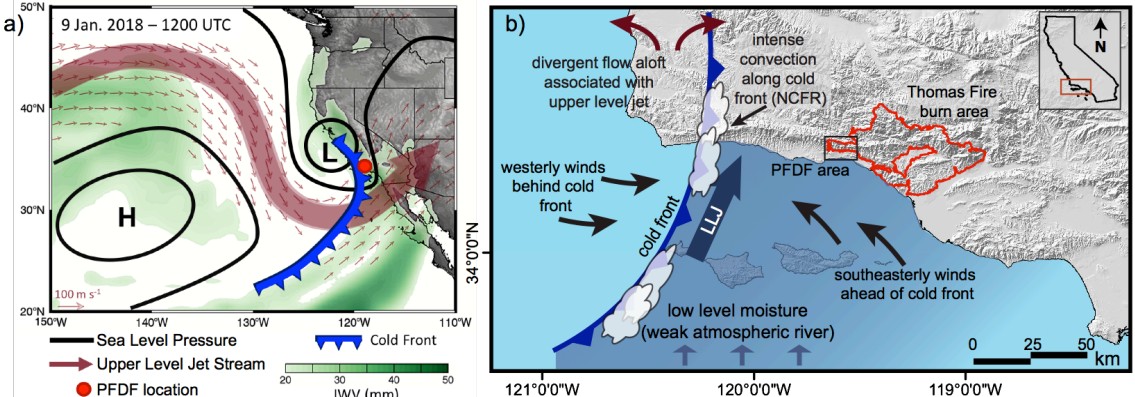

**Figure 2: a) A conceptual synoptic view of conditions at 12 UTC 9 January 2018 based on information from the Climate Forecast System version 2 operational analysis (https://www.ncdc.noaa.gov/data-access/model-data/model-datasets/climate-forecast-system-version2-cfsv2). b) A conceptual mesoscale view of conditions preceding the event as the cold front approaches the Thomas Fire burn area.**





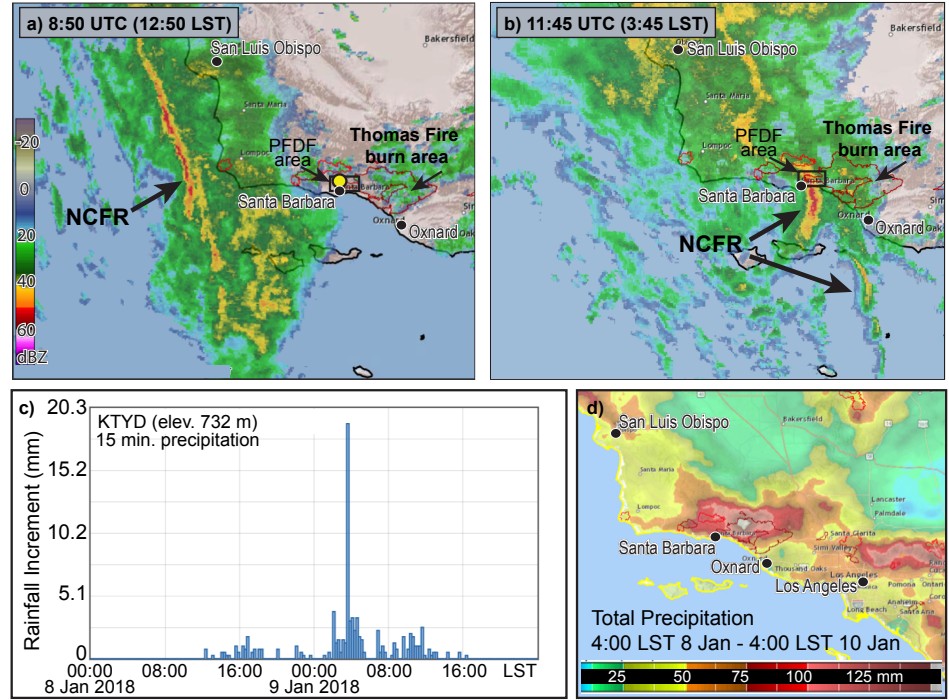

**Figure 3: Panels a and b show radar imagery a) preceding and b) at the time of PFDFs on the Thomas Fire burn area. Yellow to red colors indicate higher intensity precipitation. Radar image source: California-Nevada River Forecast Center (CNRFC; cnrfc.noaa.gov). Panel c) shows precipitation observations at 15-minute intervals from the KTYD station operated by SBCPWD. The station position is indicated by the yellow marker in panel a). Panel d) provides regional 48 h precipitation totals from CNRFC.**