# Peer review of "Brief Communication: Meteorological and climatological conditions associated with the 9 January 2018 post-fire debris flows in Montecito and Carpinteria California, USA"

_Natural Hazards and Earth System Sciences, 2018_

## Referee Comment (RC1) · Anonymous Referee #1 · 3 Jul 2018

This manuscript describes the meteorological conditions and climatological reference points (e.g., return period estimates) of the heavy rainfall that drove catastrophic debris flows following the 2017-2018 southern California wildfires. This is admittedly my first review of a "Brief Communication" submission, and in all honesty as I read it, I struggled to find novel aspects that were obviously worthy of publication. The event itself is interesting and high-impact, the data summary and meteorological analysis is sound, and the writing and communication is clear. Thus, the main issues that I have are more to do with what seems to be lacking, rather than problems with the material in the

manuscript. However, suspecting that the problem may be with my own expectations of a full-length publication relative to the present manuscript type, I offer below only a few minor comments/suggestions that the editor and authors can consider as they deem appropriate.

General comment: 1. If part of the purpose of this manuscript is to " support investigations on this and other PFDFs in a range of fields...," then I suggest adding at least some discussion of/references to relevant post-fire hydrologic or geologic concepts that might be of interest in future research, e.g., a. Neary et al. 2003: https://www.researchgate.net/publication/228510172_Post-wildfire_watershed_flood_responses b. 'Havel et al. 2018: https://doi.org/10.5194/hess-22-2527-2018 c. Brogan et al. 2017 https://onlinelibrary.wiley.com/doi/full/10.1002/esp.4194

Specific comments: Lines 25 – 26: I'm not familiar with the language/terminology "having high debris flow hazard"...do you mean risk? Can you re-phrase/explain for a general audience?

Figs. 3a, b are highly suggestive of possible line echo wave pattern ("LEWP") dynamics. Again, in the interest of supporting/inspiring future investigations, perhaps a reference to this idea/possibility be added.

---

## Referee Comment (RC2) · Anonymous Referee #2 · 27 Jul 2018

A bit puzzled on the whole process here. Not seeing any open scientific discussion having occurred at all, just the comments made weeks ago by Anonymous Referee #1. In the absence of the former, fail to see how the process of peer review and publication in Natural Hazards and Earth System Sciences (NHESS) differs from traditional scientific journals. Also unclear on what the expectations are for a "Brief Communication" submission and am unable to find information in that regard. It is with those caveats that this review is provided, and I leave it to the editor and authors as to how they wish to consider my comments. Recommendation: Accept for publication after

suitable moderate to major revision. • Major Comment #1: Would like to see this focused down to what the key triggering meteorological event was, the accompanying hydro-meteorological circumstances that resulted in the extreme outcome, and the basic synoptic and mesoscale evolution. Much of that is already there, but believe it could be better organized to present a clearer picture. o In section 2.1, just give the basic synoptic evolution – say 500 mb, SLP and IWV every 12 hours for the 36 or so hours leading up to the event. Can omit the rest of it. o Not immediately seeing the connection between this event and atmospheric rivers. Page 2, lines 20-26: (i) need to provide evidence in support of the claim that the moisture plume resulted from re-organization of the remnant moisture from the AR that moved through the previous day.  (ii) Are you really making the claim that this event itself was associated with a weak AR? Are the spatial scales consistent with the definition of an AR? And then might want to expand a bit on the consequent implication that weak ARs can potentially result in catastrophic hydro events.  On the other hand, if it isn't an AR, would be worth noting that catastrophic hydro events can occur in coastal California that are not associated with ARs. Either way, it's interesting and important, just needs to be clarified.

o In section 2.2, just need clear sequences of satellite images, radar images, and surface analyses leading up to the event. o New section 2.3: focus down on the microscale event itself, when and where the 5 to 15 minute extreme precip bursts occurred, how much fell, and in relation the exact locations and time frame of the debris flows.

• Major Comment #2: After reducing down to and organizing key figs, recommend including all in the manuscript itself rather than some as "supplemental material."

• Major Comment #3: Strongly recommend confining the focus to this event, especially given the "Brief Communication" nature of the submission (and thus eliminating Figs S10, S11 and accompanying discussion, etc)

• Other Comments: o Page 1, this event occurred on January 9 but the Thomas Fire

not 100% contained until January 12? o Page 1, might want to note how long it had been since last significant precip o Page 1, lines 28-29: cite ref re exceeding USGS 15-min design storm. . . o Page 3, line 2: Markowski and Richardson, 2010 not found in Reference section. o Page 3, line 9: intense convective precip bands? But sounding in Fig S6 shows ∼ zero CAPE. o Page 3, line 21: created o Page 4, lines 30-31: thought this NCFR developed behind the primary AR, not in it. o References: not entirely in alphabetical order.

---

## Author Comment (AC1) · 28 Sep 2018

Comments made by Anonymous Referee 1 are provided in black text.
Author responses are provided in blue text.

Anonymous Referee #1

This manuscript describes the meteorological conditions and climatological reference
points (e.g., return period estimates) of the heavy rainfall that drove catastrophic debris flows following the 2017-2018 southern California wildfires. This is admittedly my first review of a "Brief Communication" submission, and in all honesty as I read it, I struggled to find novel aspects that were obviously worthy of publication. The event itself is interesting and high-impact, the data summary and meteorological analysis is sound, and the writing and communication is clear. Thus, the main issues that I have are more to do with what seems to be lacking, rather than problems with the material in the manuscript. However, suspecting that the problem may be with my own expectations of a full-length publication relative to the present manuscript type, I offer below only a few minor comments/suggestions that the editor and authors can consider as they deem appropriate.

The reviewer is correct, it was not our goal here to answer a question or test a hypothesis, but rather to document the incident in a way that hopefully provides insight to future research. Following the event, the authors saw misinterpretation of the meteorological characteristics of the event in the media and discussions with colleagues. We read and heard references to the event being a tropical storm, or being caused by orographic effects alone. We were concerned these misinterpretations may be damaging to the progression of research on the topics of short duration, high intensity rainfall in this region and post-fire debris flows. Many scientific publications related to this event are anticipated over the next few years. A peer-reviewed statement on the meteorological drivers of this event provides researchers in a variety of fields a reference to support their work, without having meteorology expertise. A brief communication, rather than a full-length article seemed the appropriate place to make a statement on the event.

General comment: 1.
If part of the purpose of this manuscript is to " support investigations on this and other PFDFs in a range of fields...," then I suggest adding at least some discussion of/references to relevant post-fire hydrologic or geologic concepts that might be of interest in future research, e.g.,
a..Neary et al. 2003: https://www.researchgate.net/publication/228510172_Post-wildfire_watershed_flood_responses
b.'Havel et al. 2018: https://doi.org/10.5194/hess-22-2527-2018
c. Brogan et al. 2017 https://onlinelibrary.wiley.com/doi/full/10.1002/esp.4194

Our intent is to focus on the meteorological triggers of the debris flow, rather than the geomorphic processes. Additionally, we are limited to 20 references and

~2500 words (NHESS author guidelines for Brief Communications can be found here: https://www.natural-hazards-and-earth-system-sciences.net/about/manuscript_types.html), so we chose to spend these on meteorological references and references specific to the event. To provide clarification, we have adjusted the paragraph at page 2, line 10 to explain that we are supporting the understanding of the meteorological triggers of PFDFs, which has been shown to be a knowledge gap in management communities (e.g. Garfin et al. 2016). This provides differentiation from the initial statement that could have been interpreted as if we were attempting to educate on post-fire debris flow processes in general.

As the reviewer alludes to, there is a large body of work on post-fire debris flow geomorphic processes, and here we choose to focus on the meteorological triggering process for this event and relate it to past events and climatology.

Specific comments: Lines 25 – 26: I'm not familiar with the language/terminology "having high debris flow hazard"...do you mean risk? Can you re-phrase/explain for a general audience?

The USGS model is a hazard model because it does not intersect the debris flow hazard with values exposed to and vulnerable to the hazard. Therefore in this case, it is correct to say that the USGS model provides information about debris flow hazards rather than risk.

Figs. 3a, b are highly suggestive of possible line echo wave pattern ("LEWP") dynamics. Again, in the interest of supporting/inspiring future investigations, perhaps a reference to this idea/possibility be added?

Line echo wave patterns are a type of squall line (Markowski and Richardson, 2010). NCFRs can be differentiated from squall lines based on their low-level cold pool maintenance by cold-air advection (CAA) rather than evaporational cooling in shallow rear-to-front flow (Geerts and Hobbs 1995). We see evidence of CAA in backing winds in vertical wind profile observed at Santa Barbara Airport following the frontal passage (Fig. S8).

LEWPs are most commonly described as a phenomenon impacting areas east of the Rockies during the spring and summer seasons (e.g. Johns 1993; Weisman 1993; Pryzybylinski 1995; Corfidi et al. 2018). LEWPs are typically associated with extreme convective instability owing to high amounts low-level moisture (Pryzybylinski 1995). Johns et al. (1990) found that long-lived derechos had an average of 2400 J/kg of CAPE in the initiation environment. It is much less likely for LEWPs to develop in low-instability environments such as the one observed here.

Additionally, LEWPs are commonly associated with straight-line strong wind events known as derechos (Pryzybylinski 1995 and references therein). The

NWS criteria for derechos requires wind gusts >57 mph at most points along the storm path (Corfidi et al. 2018). At stations near sea level in this event, we see gusts typically <=40 mph (37.8 mph at KSBA, 24 mph at Montecito #2 RAWS, 40 mph at Santa Barbara Botanic Garden RAWS, and 28 mph at Casitas RAWS; https://raws.dri.edu/). At higher elevation stations like San Marcos Pass RAWS (457 m), wind speed was slightly higher, with a maximum of 48 mph. For all these cases, maximum gusts were out of the S/SE. The increase in wind speed with elevation is consistent with the presence of a strong low-level jet (Neiman et al. 2004). The LLJ also displays well in the vertical wind profile from KSBA (Figure S8) and a burst of wind at the surface consistent with a derecho event is not present in the profiler data. The lack of strong "downburst"-type winds weakens the case for the consideration of this event as a LEWP.

References:

Corfidi, S., Evans J., Johns, R (last updated 2018). About Derechos. https://www.spc.noaa.gov/misc/AbtDerechos/derechofacts.htm  Accessed online 31 Aug 2018.

Garfin, G., LeRoy, S., Martin, D., Hammersley, M., Youberg, A., Quay, R. (2016). *Managing for future risks of fire, extreme precipitation, and post-fire flooding. Report to the U.S. Bureau of Reclamation, from the project Enhancing Water Supply Reliability*. Tucson, AZ: Institute of the Environment. https://bit.ly/2LRhCtQ

Geerts B., P.V. Hobbs, 1995: A squall-like narrow cold-frontal rainband diagnosed by combined thermodynamic and cloud microphysical retrieval. *Atmos. Res.* **39**, 287-311.

Johns, R. H. (1990). Conditions associated with long-lived derechos-An examination of the large-scale environment. In *16th Conf. on Severe Local Storms, Kananaskis Park, Albert, Canada, Amer. Meteor. Soc., 1990* (pp. 408-412).

Johns, R.H., 1993: Meteorological Conditions Associated with Bow Echo Development in Convective Storms. *Wea. Forecasting,* **8**, 294–299, https://doi.org/10.1175/1520-0434(1993)008<0294:MCAWBE>2.0.CO;2

Przybylinski, Ron W. "The bow echo: Observations, numerical simulations, and severe weather detection methods." *Weather and Forecasting* 10.2 (1995): 203-218.

Weisman, M. L. (1993). The genesis of severe, long-lived bow echoes. *Journal of the atmospheric sciences*, *50*(4), 645-670.

---

## Author Comment (AC2) · 28 Sep 2018

Comments made by Anonymous Referee 1 are provided in black text.
Author responses are provided in blue text.

Reviewer #2 Comments

A bit puzzled on the whole process here. Not seeing any open scientific discussion having occurred at all, just the comments made weeks ago by Anonymous Referee #1.In the absence of the former, fail to see how the process of peer review and publication in Natural Hazards and Earth System Sciences (NHESS) differs from traditional scientific journals. Also unclear on what the expectations are for a "Brief Communication" submission and am unable to find information in that regard. It is with those caveats that this review is provided, and I leave it to the editor and authors as to how they wish to consider my comments. Recommendation: Accept for publication after suitable moderate to major revision.

In this discussion context, both reviewer or public comments and author responses are available to the public during review and after publication. At the beginning of the manuscript open discussion process in late June 2018, the link to the discussion was posted on the National Weather Service Los Angeles/Oxnard Facebook and Twitter pages, which have 35K and 20K followers, respectively. The same week, we sent the link to over 100 attendees of the International Atmospheric Rivers Conference as well as to a group of approximately 30 scientists and emergency managers who had attended a workshop on the Montecito debris flows in February 2018 at the University of California, Santa Barbara. Despite our efforts to make the discussion paper known and invite commentary, we did not receive any beyond the anonymous reviewer evaluations. NHESS statistics do show over 700 views and 146 downloads of the discussion paper.

Major Comment #1: Would like to see this focused down to what the key triggering meteorological event was, the accompanying hydrometeorological circumstances that resulted in the extreme outcome, and the basic synoptic and mesoscale evolution. Much of that is already there, but believe it could be better organized to present a clearer picture.

- In section 2.1, just give the basic synoptic evolution – say 500 mb, SLP and IWV every 12 hours for the 36 or so hours leading up to the event. Can omit the rest of it.

In the initial submission we were limited to three figures, ~2500 words, and 20 references. We propose the following additional figure to address the synoptic evolution of the event. We have updated the text in section 2.1 to describe this sequence.

[Figure]

Figure 2: 500 hPa geopotential heights (black contour lines), sea level pressure (pink contour lines) and integrated water vapor (IWV; green filled contours) at 6-hour intervals for 36 h preceding to the event, time nearest event (outlined in pink), and twelve hours following the event.

- Not immediately seeing the connection between this event and atmospheric rivers. Page 2, lines 20-26:
  - (i) need to provide evidence in support of the claim that the moisture plume resulted from re-organization of the remnant moisture from the AR that moved through the previous day.

Removed reference to this reorganization (to describe this process in full detail is beyond the scope of the project) and introduced figure

above to help describe the evolution of moisture plumes in this event. In lines 20-26, we now provide a simple narrative of the evolution of the moisture plumes.

- o  (ii) Are you really making the claim that this event itself was associated with a weak AR? Are the spatial scales consistent with the definition of an AR? And then might want to expand a bit on the consequent implication that weak ARs can potentially result in catastrophic hydro events
- o  On the other hand, if it isn't an AR, would be worth noting that catastrophic hydro events can occur in coastal California that are not associated with ARs. Either way, it's interesting and important, just needs to be clarified.

We do indeed interpret this as a weak AR. Restructured this section to state that both the IWV and IVT values and the shape and orientation of this moisture plume are consistent with the definition of an atmospheric river, though a weak one. Additionally, to address the important point the reviewer makes, we added to the conclusion that, "A weak atmospheric river was present at the time of the event, demonstrating that catastrophic hydrologic impacts can occur even in the absence of substantial water vapor transport (i.e., a strong atmospheric river) due to synoptic-to-mesoscale forcing."

- • In section 2.2, just need clear sequences of satellite images, radar images, and surface analyses leading up to the event.

Given the brevity of this manuscript, it is not feasible to include all variables suggested. We focus on a few key features of interest to make a short communication on some of the main features observed. We do provide satellite imagery in the supplementary material (Fig S7), radar imagery in Fig 4 and S9, and now have SLP in Fig 2 (see above) and timeseries of surface winds available in the profiler data in Fig S8.

- • New section 2.3: focus down on the microscale event itself, when and where the 5 to 15 minute extreme precip bursts occurred, how much fell, and in relation the exact locations and time frame of the debris flows.

Our intention is to show the 5-minute high intensity rainfall, its timing, and locations of the debris flows in Fig 1, with additional info in Tables S1-S3. It is well established that post-fire debris flows occur within moments of intense rainfall (e.g. Kean et al. 2011), thus the times provided can be

considered associated with debris flow occurrence. We have added additional references to section 3.1 (historical context of precipitation).

Major Comment #2: After reducing down to and organizing key figs, recommend including all in the manuscript itself rather than some as "supplemental material."

We are limited to three tables/figures in an NHESS Brief Communication, though we are hoping to include a fourth figure (that shown above) with the editor's approval. Our audience is primarily non-meteorologists, so we have chosen to focus on a few basic variables to communicate a concise message on key characteristics of the event. We want to demonstrate that the event was not caused by orographic enhancement alone, an extreme atmospheric river, or a tropical storm (all descriptions observed in the media). We include several supplementary figures to help support those who would like additional information.

Major Comment #3: Strongly recommend confining the focus to this event, especially given the "Brief Communication" nature of the submission (and thus eliminating Figs S10, S11 and accompanying discussion, etc)

One of our main goals is to put this event in context of historic events, both in terms of rainfall amounts and the meteorological conditions surrounding the event. We would like the reader to understand this was not a rare meteorological event for the area, and that mesoscale features such as NCFRs producing short duration, high intensity rainfall capable of initiating PFDFs are relatively commonplace in this region.

Other Comments:
- Page 1, this event occurred on January 9 but the Thomas Fire not 100% contained until January 12?
    - This is correct, the Thomas Fire was not declared fully contained until Jan 12. Rains on Jan 9 helped firefighters to extinguish the fire. http://www.latimes.com/local/lanow/la-me-thomas-fire-contained-20180112-story.html
- Page 1, might want to note how long it had been since last significant precip
    - This was the first significant rainfall event of the season, and this piece of info is likely of interest to readers. It has been added on page 1, ~line 27

- Page 1, lines 28-29: cite ref re exceeding USGS 15-min design storm
    - Added citation
- Page 3, line 2: Markowski and Richardson, 2010 not found in Reference section.
    - Reference was present, however, in formatting ended up tacked on to the end of the preceding reference. They have now been properly spaced.
- Page 3, line 9: intense convective precip bands? But sounding in Fig S6 shows zero CAPE.
    - This event does not feature substantial CAPE, typical for cool season events in this region. We have added to the Supplement Figure S5, which shows the sounding at the model timestep prior to the event (09 UTC) to complement Fig S6, sounding at the model timestep immediately following the event (12 UTC). In the sounding prior to the event, the most unstable parcel CAPE is 168 J/kg. At the timestep following the event, most unstable parcel CAPE is 42 J/kg. In a study of 19 historic events that produced post-fire debris flows, median CAPE is typically <50 J/kg at the time of the event; most events feature a moist-neutral profile (Oakley et al. 2017). A narrow cold frontal rainband is a line of intense (sometimes forced rather than free) convection associated with the density-current action of the low level leading edge of the cold front (Houze 2014). The documented cases of this type of rainband indicate that is can be produced by the forced ascent of stable or only slightly unstable air (Houze et al. 2014). If you get dynamical forcing in the moist neutral layer, as in this case along the cold front, you can release potential instability by moving the moist-neutral parcel to a higher elevation. We describe this process in Section 2.2, lines 10-12.

- Page 3, line 21: created
    - Made change
- Page 4, lines 30-31: thought this NCFR developed behind the primary AR, not in it.
    - Added "in association with" to clarify
- References: not entirely in alphabetical order.
    - Made change

References:

Houze Jr, R. A. (2014). *Cloud Dynamics* . Academic Press, 573 pp.

Kean, J. W., Staley, D. M., & Cannon, S. H. (2011). In situ measurements of post-fire debris flows in southern California: Comparisons of the timing and magnitude of 24 debris-flow events with rainfall and soil moisture conditions. *Journal of Geophysical Research: Earth Surface*, *116*(F4).

Oakley, N. S., Lancaster, J. T., Kaplan, M. L., & Ralph, F. M. (2017). Synoptic conditions associated with cool season post-fire debris flows in the Transverse Ranges of southern California. *Natural Hazards*, *88*(1), 327-354.

---

## Author Comment (AC3) · 28 Sep 2018

In addition to responding to reviewer comments, the revised version of this manuscript will feature an update to precipitation observations for the Doulton Tunnel weather station. Examination of the record of data transmissions during the event revealed several observations that were not transmitted properly, thus the actual observed values at this gauge were higher than the initial values provided in this manuscript and supplement.

We have also introduced 1-2 sentences describing the importance of the 15-min precipitation duration in post-fire debris flow occurrence.